# Occurrence and Genotypic Identification of *Blastocystis* spp., *Enterocytozoon bieneusi*, and *Giardia duodenalis* in Leizhou Black Goats in Zhanjiang City, Guangdong Province, China

**DOI:** 10.3390/ani13172777

**Published:** 2023-08-31

**Authors:** Xingang Yu, Hongcai Wang, Yilong Li, Xuanru Mu, Kaijian Yuan, Anfeng Wu, Jianchao Guo, Yang Hong, Haoji Zhang

**Affiliations:** 1School of Life Science and Engineering, Foshan University, Foshan 528231, China; yuxingang4525@163.com (X.Y.); whc25882021@163.com (H.W.); muylyl20001110@163.com (Y.L.); 15940815092@163.com (X.M.); yuankj0195@163.com (K.Y.); 2Maccura Biotechnology Co., Ltd., Chengdu 510000, China; 18813291567@163.com; 3Guangdong Provincial Animal Husbandry Technology Promotion Station, Guangzhou 510500, China; jianchaoguo000@163.com; 4National Institute of Parasitic Diseases, Chinese Center for Diseases Control and Prevention (Chinese Center for Tropical Diseases Research), Key Laboratory of Parasite and Vector Biology, National Health Commission of the People’s Republic of China (NHC), World Health Organization (WHO) Collaborating Center for Tropical Diseases, National Center for International Research on Tropical Diseases, Shanghai 200025, China

**Keywords:** *Blastocystis* spp., *Enterocytozoon bieneusi*, *Giardia duodenalis*, molecular epidemiology, Leizhou black goats

## Abstract

**Simple Summary:**

We report the occurrence of *Blastocystis* spp., *Enterocytozoon bieneusi*, and *Giardia duodenalis* infections in Leizhou black goats in Zhanjiang City, Guangdong Province, China. The total prevalence rates of *Blastocystis* spp., *E. bieneusi*, and *G. duodenalis* were 33.63% (76/226), 17.7% (40/226), and 24.78% (56/226), respectively. Four *Blastocystis* spp. subtypes (ST5, ST10, ST14, and ST21), four *E. bieneusi* genotypes (CHG3, CM21, CHG1, and ET-L2) and two assemblages (A and E) of *G. duodenalis* were identified. The detection of zoonotic pathogen species, genotypes, and assemblages in Leizhou black goats implied their potential involvement in the transmission of zoonotic parasitic diseases.

**Abstract:**

*Blastocystis* spp., *Enterocytozoon bieneusi*, and *Giardia duodenalis* are three common zoonotic intestinal parasites that cause severe diarrhea and enteric diseases. Leizhou black goats are characterized by a high reproductive rate, fast growth, and good meat quality, making them one of the pre-eminent goat breeds in China. Goats are reportedly common reservoirs of these three intestinal pathogens, but no information on their prevalence or genotypic distributions in black goats in Guangdong Province, China, is available. A total of 226 fecal samples were collected from goats in Zhanjiang city and genomic DNA was extracted from them. The presence of the three pathogens was detected using nested PCR targeting the sequences encoding SSU rRNA (*Blastocystis* spp.), the internal transcribed spacer of rRNA (ITS; *E. bieneusi*), as well as beta-giardin, glutamate dehydrogenase, and triosephosphate isomerase (*G. duodenalis*). All PCR products were sequenced to determine the species and genotypes of the organisms. The total prevalence rates of *Blastocystis* spp., *E. bieneusi*, and *G. duodenalis* were 33.63% (76/226), 17.70% (40/226), and 24.78% (56/226), respectively. Four subtypes of *Blastocystis* spp. were detected: ST5 (*n* = 6), ST10 (*n* = 50), ST14 (*n* = 14), and ST21 (*n* = 6). Among them, ST10 was the dominant genotype, accounting for 65.79% of strains, followed by the genotypes ST14 (18.42%), zoonotic ST5 (7.89%), and ST21 (7.89%). Four genotypes of *E. bieneusi* were detected: CHG3 (*n* = 32), CM21 (*n* = 4), CHG1 (*n* = 2), and ET-L2 (*n* = 2). Among these, CHG3 was the dominant genotype. Assemblage E (*n* = 54) and concurrent assemblages A and E (*n* = 2) were identified in the *G. duodenalis*-positive goats using multilocus genotyping. *Blastocystis* spp., *E. bieneusi*, and *G. duodenalis* infections were common in Leizhou black goats, all of which have zoonotic genotypes, indicating the potential risk of zoonotic transmission. Our results provide basic data for the prevention and control of these three intestinal pathogens. Further studies are required to better understand their genetic characteristics and zoonotic potential in Guangdong Province.

## 1. Introduction

*Blastocystis* spp., *Enterocytozoon bieneusi*, and *Giardia duodenalis* are three common opportunistic intestinal parasitic protozoa with wide host ranges, which include domestic animals, wildlife, and humans [1,2,3]. The oocysts or cysts of these three pathogens can survive in the environment for a long time. Human and animal infections often occur through fecal–oral contact or contact with contaminated water or food. These pathogens may cause severe abdominal pain, diarrhea, emaciation, and even death [2,4].

Among the approved subtypes (ST1–ST34) of *Blastocystis* species, ST1–ST10, ST12, ST14, ST16, and ST23 have been observed in humans [5]. In ASEAN countries, 11 subtypes of *Blastocystis* spp. have been identified in Artiodactyla: ST1–ST8, ST10, ST12, and ST14 [6]. In China, seven subtypes, ST1, ST3, ST4, ST5, ST6, ST10, and ST14, mainly infect ruminants, including cattle, sheep, and goats [7,8], and ST1, ST3, ST6, and ST7 have been detected in goats in Malaysia [9]. The occurrence of the zoonotic *Blastocystis* spp. subtypes ST1, ST3, ST4, ST5, ST7, and ST10 in goats suggests their potential transmission to humans.

*Enterocytozoon bieneusi* is a complex species, divided into 13 different genetic groups based on the sequence of the internal transcribed spacer (ITS) region of the rRNA genes [10]. Groups 1 and 2 contain the majority of zoonotic genotypes, whereas groups 3–13 contain host-adapted genotypes [11,12,13]. *Enterocytozoon bieneusi* has an estimated overall prevalence of 17.4% in sheep and 16.3% in goats worldwide [14]. Analyses of the ITS sequences have shown that the genotypes BEB6 and COS-1 are most common in sheep, whereas CHG3 and BEB6 are the predominant genotypes in goats [14,15].

Giardiasis has been prevalent or has occurred in outbreaks throughout the world since the 1970s, and is listed by the World Health Organization as one of the neglected diseases that endanger human health [16]. It is estimated that more than 280 million cases of human giardiasis occur annually throughout the world [17]. *Giardia duodenalis* can be divided into eight assemblages (A–H) according to the different genotypes. The zoonotic assemblages A and B and host-adapted assemblage E of *G. duodenalis* have been detected in goats, and previous reports have cited assemblage E as the dominant variant [3]. In goats, *G. duodenalis* infections can occur at any age, but are most commonly observed in young and immunocompromised animals. Genotyping *G. duodenalis* is a valuable way to identify zoonotic assemblages, and is achieved with sequence analyses of the PCR products of the glutamate dehydrogenase (*gdh*), β-giardin (*bg*), and triosephosphate isomerase genes (*tpi*) with multilocus genotyping (MLG) [18].

Goats are grain-saving types of livestock that provide high-quality meat, cashmere, and milk. These animals have an important role in animal husbandry in China. The Leizhou black goat (*Capra hircus*) and Chuanzhong black goat (*C. hircus*) are the two most widely farmed meat goat breeds in Guangdong, although the Leizhou black goat is the only landrace meat goat [19]. Zhanjiang city has the largest number of black goats in Guangdong Province. Black goats in local areas are usually free-range, so their excrement is usually deposited directly into the surrounding environment without further treatment. Three zoonotic parasites of goats, *Blastocystis* spp., *E. bieneusi*, and *G. duodenalis*, not only cause significant production losses in goats, but also pose a potential threat to public health through the cysts or oocysts excreted in goat feces. Understanding the sources and genetic diversity of *Blastocystis* spp., *E. bieneusi*, and *G. duodenalis* involved in these infections is essential to understanding their pathogenicity and the development of strategies for their control, because no effective vaccines or drugs are available. However, there is currently no information on *Blastocystis* spp., *E. bieneusi*, or *G. duodenalis* infections in Leizhou black goats.

The purpose of this study was to examine the occurrence and genotypic distributions of these three intestinal pathogens in Leizhou black goats in Zhanjiang city, in the southernmost region of Guangdong Province, China.

## 2. Materials and Methods

### 2.1. Sample Collection

Between October and December 2022, 226 fresh fecal samples from Leizhou black goats were collected on four farms scattered within Zhanjiang city, Guangdong Province (Figure 1). The samples were from 96 growing goats (aged 4–18 months), 64 reserve goats (aged 19–30 months), and 66 adult goats (aged >30 months). Each fresh fecal sample was placed in an individual disposable plastic pouch marked with the farm, goat age, and date of collection, transported quickly to the laboratory while packed in ice, and stored in 2.50% (*w*/*v*) potassium dichromate solution at 4 °C until processing (<1 week). 

### 2.2. DNA Extraction

Each fecal sample was washed with distilled water to remove the potassium dichromate before DNA extraction. Approximately 200 mg of each fecal sample was used to extract DNA with an E.Z.N.A.^®^ Stool DNA Kit (Omega Bio-tek Inc., Norcross, GA, USA), according to the instructions of the kit. The DNA samples were stored at −20 °C before PCR analysis.

### 2.3. PCR Amplification

The DNA from all the samples was amplified using nested PCR to identify those positive for any of the three pathogens of interest. *Blastocystis* spp. And *E. bieneusi* were identified based on the small subunit (SSU) rRNA gene [20] and its ITS region, respectively [21]. The detection and genotyping of *G. duodenalis* was based on the nested PCR amplification and sequence analysis of the *bg*, *gdh*, and *tpi* genes (Table 1). 

The PCRs were performed in 25 μL reaction systems: 12.5 μL of 2 × rTaq mix (TaKaRa Co., Ltd., Beijing, China), 1 μL of each primer (10 μM each), 2 μL of DNA sample, and 8.5 μL of nuclease-free deionized water. The PCR products were separated with 1.20% agarose gel electrophoresis, stained with nucleic acid dye (Biosharp, Beijing, China), and observed and recorded with the Azure™ c200 Gel Image Analysis System (Dublin, CA, USA). 

### 2.4. Sequencing and Phylogenetic Analysis

All of the final positive PCR amplicons were transferred to and sequenced by Sangon Biological Engineering Technology and Service Co., Ltd. (Songjiang, Shanghai, China). BLAST (https://blast.ncbi.nlm.nih.gov/Blast.cgi, accessed on 15 June 2023.) was used to compare each obtained sequence with the GenBank database at the National Center for Biotechnology Information (NCBI). *SSU rRNA* gene sequences of *Blastocystis* spp., *ITS* gene sequences of *E. bieneusi*, and *bg* gene sequences of *G. duodenalis* obtained (Appendix A) were compared with reference sequences acquired from the GenBank database (Appendix A), respectively. Phylogenetic trees based on the amplicon sequences of *Blastocystis* spp., *E. bieneusi*, and *G. duodenalis* were constructed with the maximum likelihood (ML) method in MEGA 7.0 (http://www.megasoftware.net/, accessed on 19 June 2023) to assess their genetic relationships. Bootstrap values were calculated with 1000 replicates. 

### 2.5. Statistical Analysis

A χ^2^ test was performed and 95% confidence intervals (CIs) were calculated using the Wald method in SPSS version 27.0 (SPSS Inc., Chicago, IL, USA) to evaluate the differences in the infection rates among different age groups. Differences were considered significant at *p* < 0.05.

## 3. Results

### 3.1. Occurrence of Blastocystis spp., E. bieneusi, and G. duodenalis 

Of the 226 specimens analyzed, 76 (33.63%; 95% CI: 27.4–39.8) were positive for *Blastocystis* spp. The prevalence of *Blastocystis* spp. was significantly lower in growing goats (22.91%, 22/96; 95% CI: 14.4–31.5) than in reserve goats (46.88%, 30/64; 95% CI: 34.3–59.4; *p* = 0.03), but there was no significant difference between the growing and adult goats (36.36%, 24/66; 95% CI: 24.4–48.3; *p* = 0.09) (Table 2). 

The overall detection rate of *E. bieneusi* was 17.70% (40/226; 95% CI: 12.7–22.7). The prevalence of *E. bieneusi* was significantly lower in adult goats (6.06%, 4/66; 95% CI: 0.1–12) than in growing goats (25.00%, 24/96; 95% CI: 16.2–33.8; *p* = 0.03), but did not differ significantly between adult goats and reserve goats (18.75%, 12/64; 95% CI: 8.9–28.6; *p* = 0.053). 

Based on the PCR detection of any of the three genetic loci of *G. duodenalis* (*bg*, *tpi*, or *gdh*), *G. duodenalis* was detected in 24.78% of samples (56/226; 95% CI: 19.1–30.5). Unlike the *E. bieneusi* infection rate, which was highest in growing goats, the prevalence of *G. duodenalis* was significantly higher in reserve goats (40.63%, 26/64; 95% CI: 28.3–53; *p* = 0.000008) and growing goats (27.08%, 26/96; 95% CI: 18–36.1; *p* = 0.0015) than in adult goats (6.06%; 4/66; 95% CI: 0.1–12). 

Viewed from the perspective of coinfection, the overall percentage of goats infected with the three pathogens was 7.08% (16/226). The percentages of goats infected with only *Blastocystis* spp., *E. bieneusi*, and *G. duodenalis* were 18.58% (42/226), 5.31% (12/226), and 7.96% (18/226), respectively. Furthermore, 11.50% (26/226) of the total goats were infected with two protozoa, with 3.54% (8/226) infected with *G. duodenalis* and *E. bieneusi*, 6.19% (14/226) with *G. duodenalis* and *Blastocystis* spp., and 1.77% (4/226) with *Blastocystis* spp. and *E. bieneusi*.

### 3.2. Distributions of Blastocystis spp. Subtypes

Based on an SSU rRNA gene sequence analysis, the 76 *Blastocystis* spp.-positive isolates were characterized into four subtypes: ST5 (*n* = 6), ST10 (*n* = 50), ST14 (*n* = 14), and ST21 (*n* = 6) (Table 2, Figure 2). ST5, a zoonotic genotype, accounted for 7.89% of the positive samples, and the sequences obtained in this study were 99% identical to the that of an isolate derived from sheep in China (ON809458.1). ST10 (*n* = 50) was the predominant subtype, and 30 of the SSU rRNA sequences shared 97–100% homology with that of an isolate derived from dairy cattle in Malaysia (MK240481), whereas the other 20 sequences shared 85–98% homology with a *Blastocystis* spp. from raccoon dogs in China (MT798805). Of the 14 *Blastocystis* spp. samples of subtype ST14, 6 shared >85% homology with a *Blastocystis* spp. (ON796560) from sheep in China, and the SSU rRNA sequences of the other 8 isolates shared >95% similarity with that of MW648930 from goats in Malaysia. ST21 (*n* = 6) was detected in growing goats (4–18 months) and reserve goats (19–30 months), and shared >85% SSU rRNA homology with ON796563 from goats in China.

### 3.3. Genotypes of E. bieneusi

In the present study, four *E. bieneusi* genotypes were identified in the goats based on their genetic divergence and their positions on the ITS-based phylogenetic tree (Table 2, Figure 3): CHG3 (*n* = 32), CM21 (*n* = 4), CHG1 (*n* = 2), and ET-L2 (*n* = 2). The predominant genotype of *E. bieneusi* in goats was CHG3 (32/40, 80.00%). Based on the phylogenetic analysis of the ITS sequences determined in this study and reference sequences downloaded from GenBank, all genotypes observed in the study were categorized as group 2. The CHG3 samples (*n* = 32) were distributed in goats of all ages and shared 95–100% ITS homology with MH822618 from goats in China. CM21 (*n* = 4) was detected in reserve goats (19–30 months) and shared 99% ITS homology with KU604931 from the golden monkey in China. CHG1 (*n* = 2) was only detected in adult goats (>30 months) and shared 95% ITS homology with MH822617 from goats in China. In contrast, ET-L2 (*n* = 2) was only observed in growing goats and shared 98.6% ITS homology with MT231509 from dairy cattle in Ethiopia. 

### 3.4. Genotypes and Subtypes of G. duodenalis

Fifty-six DNA samples were positive for *G. duodenalis* according to the results for at least one gene locus (*bg*, *gdh*, or *tpi*). A total of 42, 42, and 22 samples were positive for *G. duodenalis* based on the *bg*, *gdh*, and *tpi* sequences, respectively. Two *Giardia* assemblages, E and A, were detected in these samples (Figure 4, Table 2). Two of the *bg* sequences were identified as assemblage A, and the other 40 were identified as assemblage E or subtype E12 (Appendix A). Assemblage A (*n* = 2) shared 100% *bg* homology with KP687765 from humans in Spain. Thirty *bg* sequences of assemblage E shared 97–100% homology with that of MK452880 (*n* = 30) from sheep in Greece, whereas another ten *bg* sequences shared more than 99% homology with that of KY432834 (*n* = 10) from dairy cattle in China (Figure 4). 

At the *gdh* locus, all positive samples were identified as assemblage E (*n* = 42), including subtypes E34 (*n* = 20) and E36 (*n* = 22), and shared 98% *gdh* homology with a *G. duodenalis* isolate derived from Tan sheep in China (MK645786, MK645797). Twenty-two isolates positive for *G. duodenalis* at the *tpi* locus were identified as assemblage E and shared 98–100% homology with KT922262 from calves in Ethiopia. 

## 4. Discussion

*Blastocystis* spp. is a common intestinal parasite that infects various animals worldwide, including humans, sheep, goats, birds, and insects [6,25]. Because the host specificity of *Blastocystis* spp. is low, many animals act as storage hosts for its transmission. Nineteen blastocyst subtypes (ST1–ST10, ST12–ST14, ST17–ST22) have been identified in animal populations in China, and new subtypes are still being discovered in different animal populations [26]. Among these, ST3 is the most common subtype of human infections in China, whereas ST5, ST10, and ST1 are the dominant subtypes of infections in pigs, herbivorous animals (cattle, sheep), and carnivores, respectively. The prevalence of *Blastocystis* spp. in sheep and goats ranges from 0.3% to 94.7% worldwide (after studies with small sample sizes were excluded) [27]. In China, the *Blastocystis* spp.-positivity rates in sheep and goats range from 0.3% to 58.0%, and the dominant subtype is ST10. However, there are significant differences in the subtypes of *Blastocystis* spp. in sheep and goats from different provinces or regions in China [27]. The *Blastocystis* spp.-positive rate in Leizhou black goats was 33.63% in the present study, higher than in goats in most other provinces of China, except Shaanxi (58%), but including Jiangsu (24.0%), Shandong (16.7%), Inner Mongolia (10.70%), Qinghai (7.5%), Heilongjiang (5.5%), and Anhui (0.3%) [27,28]. The prevalence of *Blastocystis* spp. in Leizhou black goats in Zhanjiang was also higher than in many other countries, including Malaysia (30.9%) and Liberia (10.5%) [27,28]. In the present study, the infection rate in growing goats was 22.91%, significantly lower than the rate in reserve goats (19–30 months; 46.88%). However, there was no significant difference between the rates of *Blastocystis* spp. infection in growing and adult goats (36.36%). 

Based on genetic typing and a phylogenetic analysis of the SSU rRNA gene sequence, four subtypes of *Blastocystis* spp. were identified: ST10 (*n* = 50), ST14 (*n* = 14), ST5 (*n* = 6), and ST21 (*n* = 6). Of these, ST10 was the predominant subtype, accounting for 65.79% of infections, consistent with the results of most studies in China [6,8,28]. Interestingly, a rarely detected subtype, ST21 (*n* = 6), was detected in the present study. The ST21 subtype has not been detected in sheep or goats in other regions of China, except recently in Tibetan sheep and Inner Mongolian goats [27,28]. We also detected the zoonotic ST5 subtype, which accounted for 7.9% (6/76) of the positive samples. ST5 is primarily found in Artiodactyla, such as pigs, goats, sheep, and cows, and in rodents, such as *Rhizomys sinensis* [29], *Callosciurus erythraeus* [30], *Hydrochoerys hydrochaeris* [31], and *Clethrionomys glareolus* [32]. In Australia, pig herds and workers in close contact with pigs showed a high prevalence of the ST5 subtype, indicating its potential transmission between animals and humans [33]. *Blastocystis* spp. has two life stages: the cyst and the trophozoite. The host often becomes infected by consuming food or drinking water contaminated with cysts. Some infected humans, especially those with mixed infections of multiple pathogens or an impaired immune system, often experience symptoms such as abdominal pain, diarrhea, and vomiting, which can be life-threatening in severe cases. Apart from zoonotic subtype ST5, it remains unclear whether any other zoonotic subtypes are present in Guangdong black goats. Because black goats potentially play a role in the transmission of *Blastocystis* spp. to humans [6], it is important that further epidemiological studies of *Blastocystis* spp. are conducted among animal husbandry workers, water sources, and black goat populations in nearby regions. 

*Enterocytozoon bieneusi* is an emerging zoonotic intestinal pathogen that infects humans and many animal species. Many studies have identified and genotyped *E. bieneusi* in black goats throughout the world [34,35,36,37]. The infection rate of *E. bieneusi* in goats ranges from 0 to 100% worldwide, and the overall prevalence in goats is estimated to be 16.3% [14]. However, genotypes and prevalence rates probably also differ by region. This is the first report of *E. bieneusi* in black goats in Guangdong Province, with a prevalence of 17.7% (40/226), which is lower than in Egypt (100%, 11/11) [37], Thailand (19.2%, 14/73) [38], Henan (66.7%, 104/156; 50.7%, 73/144; 32.9%, 113/343) [15,39,40], Chongqing (62.5%, 5/8) [40], Shaanxi (47.8%, 22/46; 21.9%, 106/485) [15,40], the Ningxia Autonomous Region (29.7%, 89/300) [41], Yunnan (22.4%, 30/134) [40], Heilongjiang (21.8%, 12/55) [42], and Qinghai (18.6%, 11/59) [43], but higher than in Slovakia (0%, 0/20) [44], Shandong (0%, 0/24) [45], Jiangsu (2.7%, 2/74) [45], Anhui (5.2%, 30/574; 7.5%, 6/80) [40,45], Yunnan (8.93%, 30/336; 10.3%, 93/907) [34,36], and the Tibet Autonomous Region (9.6%, 25/260) [46]. These results indicate the importance of goats as hosts for *E. bieneusi* and their potential role in transmitting microsporidiosis caused by this parasite.

The molecular characterization identified four *E. bieneusi* genotypes: CHG3 (*n* = 32), CM21 (*n* = 4), CHG1 (*n* = 2), and ET-L2 (*n* = 2). CHG3 was the dominant genotype, accounting for 80% of the positive samples, consistent with the results of Wang et al. [39]. CHG3 is widely distributed globally, and is found extensively in domestic animals, such as goats and sheep, in different regions or provinces of China, including in ruminant animals in northwest China [47], sheep and goats in east–central China [45], black-boned sheep and goats in Yunnan Province in southwestern China [36], and black goats in the southernmost Hainan Province [35]. However, it is interesting to note that the BEB6 genotype, which was previously reported as prevalent or present in sheep and goats in the northwest region [47], east–central China [45], and Yunnan Province [39], was not detected in the present study. The BEB6 genotype was initially considered to be specific to cattle, but is now recognized as a dominant genotype with a wide geographic distribution, a broad host range, and zoonotic potential [45]. This discrepancy may be related to the fact that our study focused primarily on goats, and previous studies have suggested that CHG3 is the dominant genotype in goats, whereas the BEB6 genotype tends to be dominant in sheep [14,39,45]. 

*G. duodenalis* is widely distributed in ruminant populations, such as goats and sheep, worldwide. The total prevalence of giardiasis in sheep and goats in China is estimated to be 7.00%, although it ranges from 0.00% to 28.93% in different provinces [48]. The prevalence of *G. duodenalis* in Leizhou black goats in Guangdong in the present study was 24.78% (56/226), higher than in goats or sheep in Anhui (1.62%, 27/1847), Gansu (1.69%, 3/177), Qinghai (2.88%, 77/1321), Henan (3.4%, 192/4434), Shaanxi (4.72%, 227/2930), Xinjiang (7.55%, 24/318), Yunnan (8.09%, 125/1568), Hainan (11%, 11/100), Sichuan (13.14%, 65/465), Ningxia (14.5%, 147/1014), and Chongqing (23.59%, 71/301), but lower than Inner Mongolia (28.93%, 150/584) [48]. Many factors may contribute to these differences: the study sample size, local climate, detection method, study design, age of the tested hosts, and the management of the animals [48]. For example, the prevalence in reserve goats (19–30 months) in the present study was 40.63% (26/64), significantly higher than in adult sheep (6.06%, 4/66). 

*G. duodenalis* strains infecting goats or sheep belong predominantly to assemblage E, but often occur in mixed infections of assemblages A and E, assemblages A and B, or assemblages A, B, and E [48,49,50,51]. Consistent with previous research [48], assemblage E was the predominant assemblage (96.43%, 54/56) in our Leizhou black goats. The co-occurrence of assemblages A and E accounted for 3.57% of infections. Assemblage A has a broad host range, infecting various mammals, including humans, domestic animals, and wildlife [52]. Assemblage E is commonly found in ruminant animals, such as sheep, goats, and cattle, but there have been increasing reports of human infections in recent years [53,54]. Preliminary investigations of the *G. duodenalis* infection status of pig and cattle herds in different regions of Guangdong Province were conducted by our laboratory [55,56]. These studies found that *G. duodenalis* infection was relatively common in pigs (18.04%, 94/521) and cattle (18.85%, 69/366) in different regions. Interestingly, the predominant assemblages of *G. duodenalis* found in cattle and pig herds are assemblage A, assemblage E, and their corresponding subtypes, whereas assemblage B was not detected in any of these livestock populations. Most parts of Guangdong Province have a subtropical monsoonal climate with abundant rainfall and a dense river network, and both assemblages (A and E), which are detected in pigs, dairy cattle and goats, possess the ability to transmit to humans and animals across species. This suggests a potential transmission cycle between workers who are in close contact with both goats and other mammals. More epidemiological research is required to better understand the spread of *G. duodenalis* among neighboring populations and water sources. 

## 5. Conclusions

This study evaluated the prevalence and genotypic characteristics of *Blastocystis* spp., *E. bieneusi*, and *G. duodenalis* in Leizhou black goats in Zhanjiang City, Guangdong Province, China. The total prevalence rates of *Blastocystis* spp., *E. bieneusi*, and *G. duodenalis* were 33.63% (76/226), 17.7% (40/226), and 24.78% (56/226), respectively. Four *Blastocystis* spp. subtypes (ST5, ST10, ST14, and ST21), four *E. bieneusi* genotypes (CHG3, CM21, CHG1, and ET-L2), and two assemblages (A and E) of *G. duodenalis* were detected. Our findings provided valuable data for understanding the molecular epidemiology of *Blastocystis* spp., *E. bieneusi*, and *G. duodenalis* in Leizhou black goats. Further research is required to understand the epidemiology and genotypic characteristics of these pathogens in animal husbandry workers and water sources in nearby regions, and the possible repercussions for public health.

## Figures and Tables

**Figure 1 animals-13-02777-f001:**
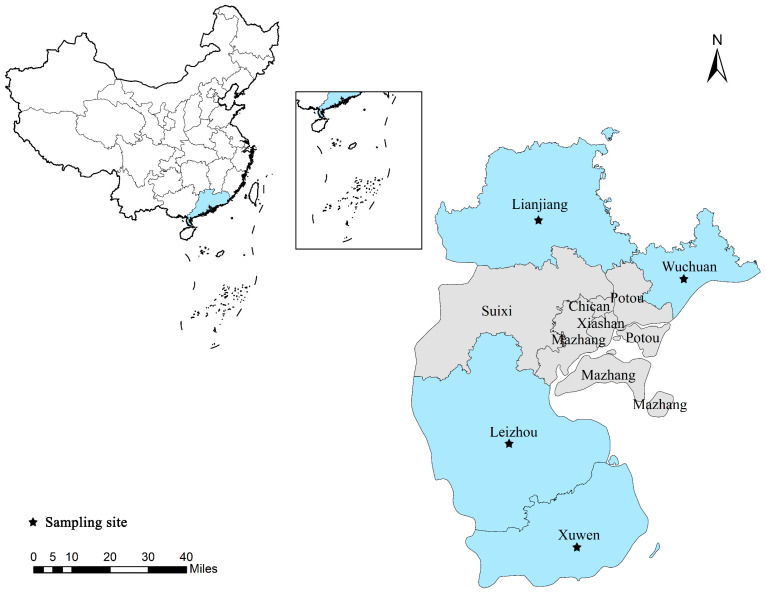
Sampling sites in the study area in Zhanjiang City, Guangdong Province, China. Black solid pentagrams indicate sampling sites.

**Figure 2 animals-13-02777-f002:**
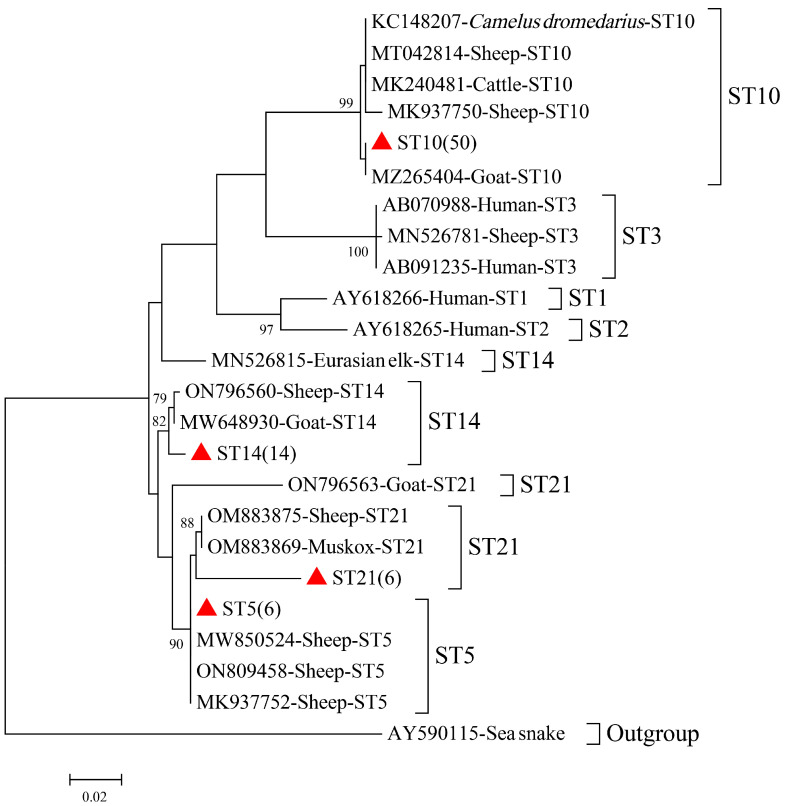
Phylogenetic tree of *Blastocystis* spp. in black goats based on *SSU rRNA* gene sequences. The Tamura–Nei model method was used with bootstrap evaluation of 1000 replicates. All the genotypes identified in this study are marked by red solid triangles. Bootstrap values are shown when >70%.

**Figure 3 animals-13-02777-f003:**
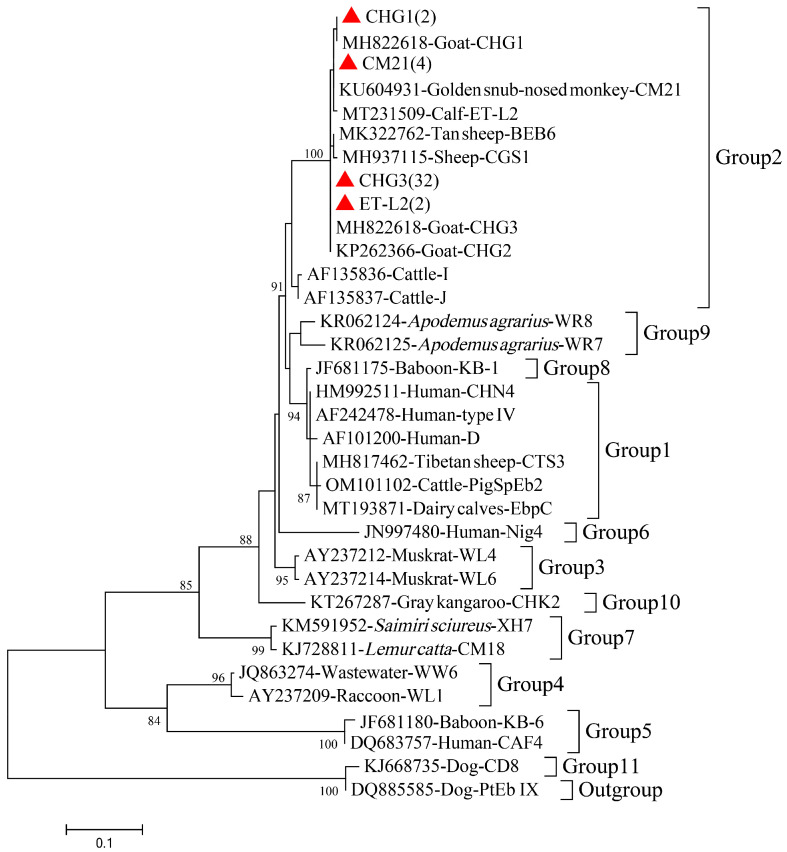
Phylogenetic tree of *E. bieneusi* in black goats based on ITS gene sequences. The Tamura–Nei model method was used with bootstrap evaluation of 1000 replicates. All the genotypes identified in this study are marked by red solid triangles. Bootstrap values are shown when >70%.

**Figure 4 animals-13-02777-f004:**
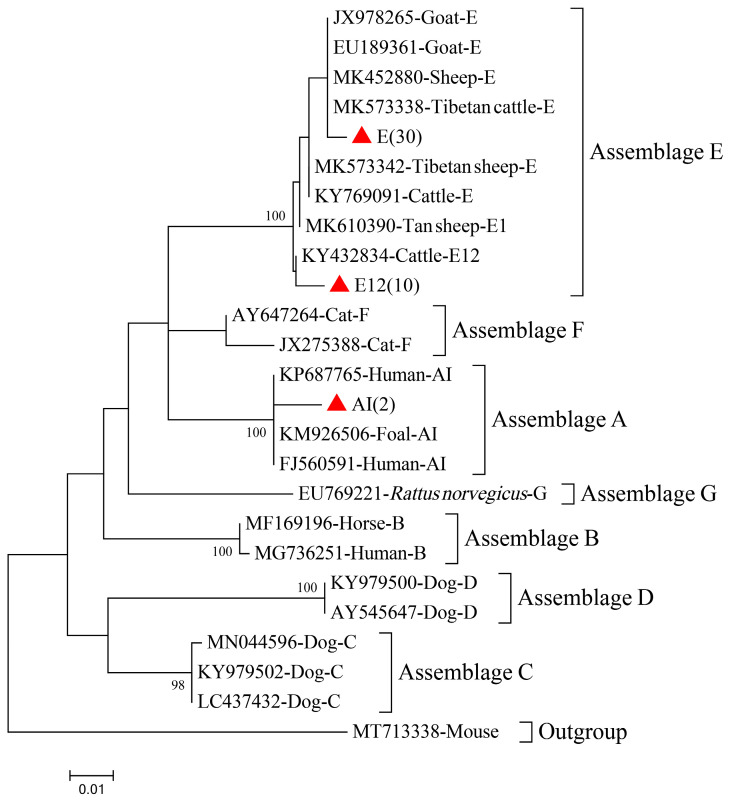
Phylogenetic tree of *G. duodenalis* in black goats based on *bg* gene sequences. The Tamura–Nei model method was used with bootstrap evaluation of 1000 replicates. All the known genotypes identified in this study are marked by red solid triangles. Bootstrap values are shown when >70%.

**Table 1 animals-13-02777-t001:** Primers used in the characterization of the *G. duodenalis*, *E. bieneusi*, and *Blastocystis* spp.

Gene	Nucleotide Sequences of Primer (5′-3′)	Expected Product Size (bp)	Annealing Temperature (°C)	Reference
*bg* gene of *G. duodenalis*	BG1: AAGCCCGACGACCTCACCCGCAGTGC	515	55	[22]
	BG2: GAGGCCGCCCTGGATCTTCGAGACGAC			
	BG3: GAACGAACGAGATCGAGGTCCG		55	
	BG4: CTCGACGAGCTTCGTGTT			
*tpi* gene of *G. duodenalis*	TPI1:AAATYATGCCTGCTCGTCG	530	57	[23]
	TPI2:CAAACCTTYTCCGCAAACC			
	TPI3:CCCTTCATCGGYGGTAACTT		57	
	TPI4:GTGGCCACCACYCCCGTGCC			
*gdh* gene of *G. duodenalis*	Gdh1: TTCCGTRTYCAGTACAACTC	530	59	[24]
	Gdh2: ACCTCGTTCTGRGTGGCGCA			
	Gdh3: ATGACYGAGCTYCAGAGGCACGT		59	
	Gdh4: GTGGCGCARGGCATGATGCA			
*ITS* gene of *E*. *bieneusi*	ITS1: GATGGTCATAGGGATGAAGAGCTT	392	57	[21]
	ITS2: TATGCTTAAGTCCAGGGAG			
	ITS3: AGGGATGAAGAGCTTCGGCTCTG		55	
	ITS4: AGTGATCCTGTATTAGGGATATT			
*SSU rRNA* of *Blastocystis* spp.	SSU rRNAF: ATCTGGTTGATCCTGCCAGT	600	55	[20]
	SSU rRNAR: GAGCTTTTTAACTGCAACAACG			

**Table 2 animals-13-02777-t002:** Colonization frequency and genotypes of *Blastocystis* spp., *E. bieneusi* and *G. duodenalis* in different age groups.

Age (Months)	Sample Size (*n*)	*Blastocystis*			*E. bieneusi*			*G. duodenalis*		
		No. Positive	Prevalence % (95% CI)	Subtypes (*n*)	No. Positive	Prevalence % (95% CI)	Genotype (*n*)	No. Positive	Prevalence% (95% CI)	Assemblage (*n*)
Growing goats (4–18 months)	96	22	22.91% (14.4–31.5)	ST10 (12) ST14 (8) ST21 (2)	24	25.00% (16.2–33.8)	CHG3 (22) ET-L2 (2)	26	27.08% (18–36.1)	E (26)
Reserve goats (19–30 months)	64	30	46.88% (34.3–59.4)	ST10 (20) ST14 (2) ST21 (4) ST5 (4)	12	18.75% (8.9–28.6)	CHG3 (8) CM21 (4)	26	40.63% (28.3–53)	E (24) A + E (2)
Adult goats (>30 months)	66	24	36.36% (24.4–48.3)	ST10 (18) ST14 (4) ST5 (2)	4	6.06% (0.1–12)	CHG3 (2) CHG1 (2)	4	6.06% (0.1–12)	E (4)
Total	226	76	33.63% (27.4–39.8)	ST5 (6) ST10 (50) ST14 (14) ST21 (6)	40	17.70% (12.7–22.7)	CHG3 (32) CHG1 (2) CM21 (4) ET-L2 (2)	56	24.78% (19.1–30.5)	E (54) A + E (2)

## Data Availability

All datasets are contained within manuscript.

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
