# Peer review of "Occurrence and Genotypic Identification of Blastocystis spp., Enterocytozoon bieneusi, and Giardia duodenalis in Leizhou Black Goats in Zhanjiang City, Guangdong Province, China"

_animals, 2023, doi:10.3390/ani13172777_

Round 1

Reviewer 1 Report

The proposed work is very interesting, as it concerns a host species potentially vehicle of zoonotic transmission for the three pathogens analyzed. It is suggested to better control the text for several errors. In particular:

- Paragraph 1, line 108: please, correct mouths with months.

- Table 1, please correct in the "Gene" column "G. dusodenalis".

- In Table 1 will be better and more helpful for the reader to report the proper primers names for each gene, according to literature. 

- In Paragraph 3: sometimes it is used Blastocystis sp. and sometimes Blastocystis spp. I suggest to homogenized it, remembering you that Blastocystis species have traditionally been defined by the type of host they live in. Therefore, all humans Blastocystis were assigned to Blastocystis hominis. According to the revision of its nomenclature, based on genetic evidence, the mammalian/avian Blastocystis isolates are now subdivided into subtypes (STs). Blastocystis from reptiles, amphibia and invertebrates maintain Linnean binomial names for the most part. 

- In Table 2 please, correct G.duodenalis in the Table title.

- In Table 2 please, correct oats with goats in the "Age" column.

- Figure 2: : MZ265404 is labelled as Got instead of Goat; KC148207 should be written in italics and as Camelus dromedarius (not Canelus). This is valid for all the scientific binomial nomenclature used in the phylogenetic trees in this paper.

Discussion section starts with an evident typo (lines 228-231). 

English should be double checked and improved in all text. 

Reviewer 2 Report

In this study, Yu et al. give important information about the prevalence of Blastocystis, Enterocytozoon bieneusi, and Giardia duodenalis, which they identified in 226 goats, as well as information about their phylogenetic characteristic. In general, the manuscript is understandable, however, it is considered that it needs some corrections.

My suggestions about the article are listed below.

Line 40. Please “When giving percentage values in the manuscript, use either one digit or two digits after the comma.” “whereas E. bieneusi is %17,7 and G. duodenalis is %24,78.”

Line 58-59. Chage to “Human and animal infections often occur through fecal-oral contact or contact with contaminated water or food. These pathogens may cause severe abdominal pain, diarrhea, emaciation, and even death.”

Line 62. Please change “Blastocystis” to “Blastocystis spp.” Please follow and implement this throughout the manuscript.

Line 86-87. Change to “Goats are grain-saving types of livestock that provide high-quality meat, cashmere, and milk. These animals have an important role in animal husbandry in China.”

Line 87. Please add the scientific names of “Leizhou black goat” and “Chuanzhong black goat”

Line 139. Did this study use the raw nucleotide data after DNA sequence analysis or the consensus sequences obtained after the sequences were processed? This part should be clarified.

Line 140. Please add parameter models for each pathogen and gene used in the ML analysis.

Line 143. Please change “1000” to “1,000” Please follow and implement this throughout the manuscript.

Line 144. Authors should supply accession numbers for their sequence from GenBank or another database. These accession numbers should add to material-methods parts of the manuscript.

Line 240 Please add reference end of the sentence “The prevalence of Blastocystis in sheep and goats ranges from 0.3% to 94.7% worldwide (after studies with small sample sizes were excluded).”

Line 275-287. In this section, the authors only provided information about the prevalence found in this study and the prevalence found in other studies. In this form, the paragraph does not provide any information to the reader. Therefore, a concluding sentence should be written in this section.

Line 304 Giardia” to “Giardia spp.” Please follow and implement this throughout the manuscript.

Line343-344. “Four Blastocystis sequence types…..” change to “Four Blastocystis sub-types…….”

best regards...

Reviewer 3 Report

1.       Why The paper is interesting for the research working on those parasites in goat.

     Why author choose protozoan and microsporidia for their research work? Coccidian are the common protozoan parasites that infect the goats. Why does the author not include it?

2.       Why author collected 226 samples? Do authors use to calculate sample size? It needs to clarify.

3.       The information of Map is not clear. Where are the study areas? Better to clear it.

4.       How about the co-infection of the parasites? The authors did not mention this in the manuscript.

5.       Pages 228-231---These lines are the comments from the reviewer. Authors need to carefully check their manuscripts before submitting them to the journal. The author needs to remove these parts and provide the response in the manuscript.

The English language is ok for me.

Reviewer 4 Report

Dear Authors

I agree with this study and offer some corrections to improve its quality as follows:

1. Some words such as "Giardia" should be written as "G. duodenalis" from the beginning to the end of the manuscript without exception; and also Blastocystis as "Blastocystis spp.".

2. In the results section, some sentences do not have proper grammatical structure and need corrections, which are highlighted in the PDF file.

3. In the result section, in addition to the results of descriptive statistics, the results of the analytical statistics including confidence intervals and p-value for each proportion should be given in "Table 2", and if there is not enough space in the "table 2", the results of different pathogens should be given in the separate tables (in addition to the text, each table should include the results of descriptive and analytical statistics related to each pathogen).

4. In the discussion section, the first paragraph should be deleted.
